# Ten years of graduates: A cross-sectional study of the practice location of doctors trained at a socially accountable medical school

John C. Hogenbirk[1]*, Roger P. Strasser[2¤], Margaret G. French[1]

1 Centre for Rural and Northern Health Research, Laurentian University, Greater Sudbury, Ontario, Canada,
2 NOSM University (Formerly the Northern Ontario School of Medicine), Greater Sudbury, Ontario, Canada

¤ Current address: Te Huataki Waiora School of Health, University of Waikato, Hamilton, New Zealand
* jhogenbirk@laurentian.ca

## Abstract

**Data Availability Statement:** Conditions of ethical approval prohibit unauthorized sharing of the full record-level survey data. Table 1 and S2–S4 Tables

### Introduction

The study predicted practice location of doctors trained at a socially accountable medical school with education programs in over 90 communities.

### Methods

A cross-sectional study examined practice location 10 years after the first class graduated from the Northern Ontario School of Medicine (NOSM), Canada. Exact tests and logistic regression models were used to assess practice location in northern Ontario; northern Canada; or other region; and rural (population <10,000) or urban community.

### Results

There were 435 doctors with 334 (77%) practising as family doctors (FPs), 62 (14%) as generalist specialists and 39 (9%) as other medical or surgical specialists. Approximately 92% (128/139) of FPs who completed both UG and PG at NOSM practised in northern Ontario in 2019, compared with 63% (43/68) who completed only their PG at NOSM, and 24% (30/127) who completed only their UG at NOSM. Overall, 37% (23/62) of generalist specialists and 23% (9/39) of other specialists practised in northern Ontario. Approximately 28% (93/334) of FPs practised in rural Canada compared with 4% (4/101) of all other specialists. FP northern Ontario practice was predicted by completing UG and PG at NOSM (adjusted odds ratio = 46, 95% confidence interval = 20–103) or completing only PG at NOSM (15, 6.0–38) relative to completing only UG at NOSM, and having a northern Ontario hometown (5.3, 2.3–12). Rural Canada practice was predicted by rural hometown (2.3, 1.3–3.8), completing only a NOSM PG (2.0, 1.0–3.9), and age (1.4, 1.1–1.8).

provide all of the aggregated data used in the study. A Minimial Data Set is available from Borealis, the Canadian Dataverse Repository (formerly Scholars Portal Dataverse) https://doi.org/10.5683/SP3/70LGLA.

**Funding:** The study was funded by the Northern Ontario School of Medicine with monies set aside from the Ontario Ministry of Health and Long-Term Care.

**Competing interests:** JCH works part time for Northern Ontario School of Medicine (now known as NOSM University) as a scholarly activity tutor in the Family Medicine program. RPS is the Founding Dean Emeritus of NOSM U. MGF worked for Centre for Rural and Northern Health Research-Laurentian during preparation of the manuscript and since September 2020 has worked full time for NOSM U as an Analyst in the Office of Institutional Intelligence. This does not alter the authors' adherence to PLOS ONE policies on sharing data

## Conclusion

This study uniquely demonstrated the interaction of two mechanisms by which medical schools can increase the proportion of doctors' practices located in economically deprived regions: first, admit medical students who grow up in the region; and second, provide immersive UG and PG medical education in the region. Both mechanisms have enabled the majority of NOSM-trained doctors to practise in the underserved region of northern Ontario.

## Introduction

The Northern Ontario School of Medicine (NOSM, renamed NOSM University in April 2022) was established twenty years ago in 2002 to be socially accountable to the people of northern Ontario [1, 2]. NOSM adopted the World Health Organization's (WHO) definition of social accountability, which is "the obligation [of medical schools] to direct their education, research and service activities towards addressing the priority health concerns of the community, region and the nation that they have a mandate to serve. The priority health concerns are to be identified jointly by governments, health-care organizations, health professionals and the public." [3, 4] To help achieve its social accountability mandate, NOSM selects students from underserved and economically deprived regions of Canada with a focus on northern Ontario. For instance, an analysis of 10 years of admissions data found that 90% of matriculating students at NOSM are from northern regions in Canada and 37% are from rural communities [5]. NOSM also provides undergraduate (UG) and postgraduate (PG) medical education in over 90 communities located primarily in the underserved and economically deprived region of northern Ontario [4]. These initiatives were designed to encourage and enable these doctors to establish their practice in the region.

In 2009, the NOSM medical student charter class graduated from NOSM's Distributed Community Engaged Learning (DCEL) program that incorporates a life-cycle approach from admission to medical school, through UG and PG medical education, and into continuing professional development [1, 2]. Key attributes of the approach include an admissions process that selects, from a pool of highly qualified candidates, medical students who reflect the population distribution of northern Ontario [5], and Immersive Community Engaged Education including the principal clinical year (clerkship year three) during which students are living in one community for eight months and learning their core clinical medicine from a community family practice perspective [6, 7]. Medical students, as well as postgraduate residents, learn and train in over 90 communities, with most of these communities scattered across the vast geography of northern Ontario [6, 7].

NOSM's selection process, UG and PG medical education, and support for continuing education align with the Rural Generalist Pathway approach [8], which is designed for economically deprived regions such as northern Ontario. NOSM's approach, which began with the admission of the charter class in 2005, is an early working example of all five health workforce education recommendations on recruitment and retention in rural and remote areas released by the WHO 16 years later in 2021 [9].

NOSM's approach explicitly demonstrates how medical schools can select students and deliver UG and PG medical education in economically deprived areas—an approach that continues to resonate around the world [10]. NOSM's education pathway, from selection, education and into practice is designed to enable NOSM-trained physicians to set up practice in the region in which they grew up and in which they trained. NOSM's approach may help weaken

the inverse care law, which states that "[t]he availability of good medical care tends to vary inversely with the need for it in the population served." [11]

NOSM's service region comprises 807,000 km$^2$ and 860,000 people, representing 89% of the land area but only 6% of the population of Ontario, respectively [12]. Approximately 40% of the population lives in communities of less than 10,000 people. An 11.5 hour drive separates the main urban centres of Greater Sudbury (162,000 people) and Thunder Bay (108,000 people), which provide tertiary care for northeast and northwest Ontario. Access to quaternary care requires an additional drive of 4–8 hours. The service area has a higher percentage of cultural-linguistic minorities relative to the province, notably Indigenous people (14% versus 2%) and Francophones (18% versus 5%) [13–15]. In comparison with all of Ontario, people living in NOSM's service region have poorer access to healthcare services, and poorer health status [16, 17]. Minority Indigenous and Francophone populations typically have worse access and worse health status [18, 19].

Previous research examined practice location of the first three and seven cohorts of family practitioners (FPs) who completed their UG, PG, or both at NOSM [20–23]. The current study extends the work to include eight cohorts of FPs and, for the first time, up to six cohorts of other specialists. This study examined the utility of demographic factors, medical education path, and medical specialty group in predicting northern Ontario practice location or rural Canada practice location of doctors who complete their undergraduate medical education, postgraduate residency training, or both at NOSM.

## Methods

In Canada, students complete an university Bachelors degree before applying to medical school [Route V in reference 24]. Upon acceptance, students complete 3–4 years of undergraduate medical education (UG) and graduate with a medical degree. In the final year of their medical degree, students apply for a postgraduate (PG) residency training position at a few medical schools. Some Canadian Medical Graduates (CMGs) go to other countries and, conversely, some International Medical Graduates (IMGs) come to Canada for their residency training positions. The number of undergraduates is fixed by government policy for each year and each school, as is the number of postgraduate residents by specialty. From 2012–2021, 41%– 46% of the residency positions were allocated to family medicine [25].

From 2005–2019, the Centre for Rural and Northern Health Research (CRaNHR) tracked medical learners who undertake UG or PG medical education or both at NOSM. CRaNHR conducted surveys during UG and PG to collect sociodemographic data [20], which were supplemented by administrative records (described below) to help mitigate bias in survey self-reporting.

CRaNHR researchers developed the research tools and methods with the advice and support of NOSM's senior leadership team. Patients and the public were and continue to be directly involved in designing NOSM's distributed, community engaged programs and in directing NOSM's strategic plan as it strives to be responsive to the needs of the service region and "engage stakeholders at all levels of its broad community." [1, 2, 4] The primary outcomes of this study were those identified by community members, though the public was not involved in the research process per se. Research Ethics Boards at Laurentian (#6020443) and Lakehead Universities (#1467348) provided ethical approval for the study. Survey respondents provided informed consent in written or electronic format. All medical learners who were invited to complete a survey were adults and were at least 20 years of age.

This cross-sectional study included all doctors who started UG in 2005 or later, completed their PG, and had some or all of their medical education or postgraduate training at NOSM.

Prior to analysis, we removed: (1) international medical graduates (IMGs) because IMGs were only at NOSM for their postgraduate training and lack a comparison groups; (2) doctors who were in practice for less than one year because some take time off, pursue extra training, or undertake locum tenens such that practice location in the first year of fully qualified practice may be in flux; and (3) doctors who only came to NOSM for their Family Medicine PG year 3 (PGY3) because we focused on full UG or full PG exposure.

The primary outcomes were a primary practice location in NOSM's northern Ontario service region (Fig 1) or in rural Canada. Rural Canada was defined by Statistics Canada as a census subdivision (CSD) of less than 10,000 people that was located outside of Census Metropolitan Area or Census Agglomeration [26]. Urban Canada was defined as a census subdivision located in a census metropolitan area or in a census agglomeration. The 2001 and 2016 censuses were used to categorise hometown and practice location, respectively.

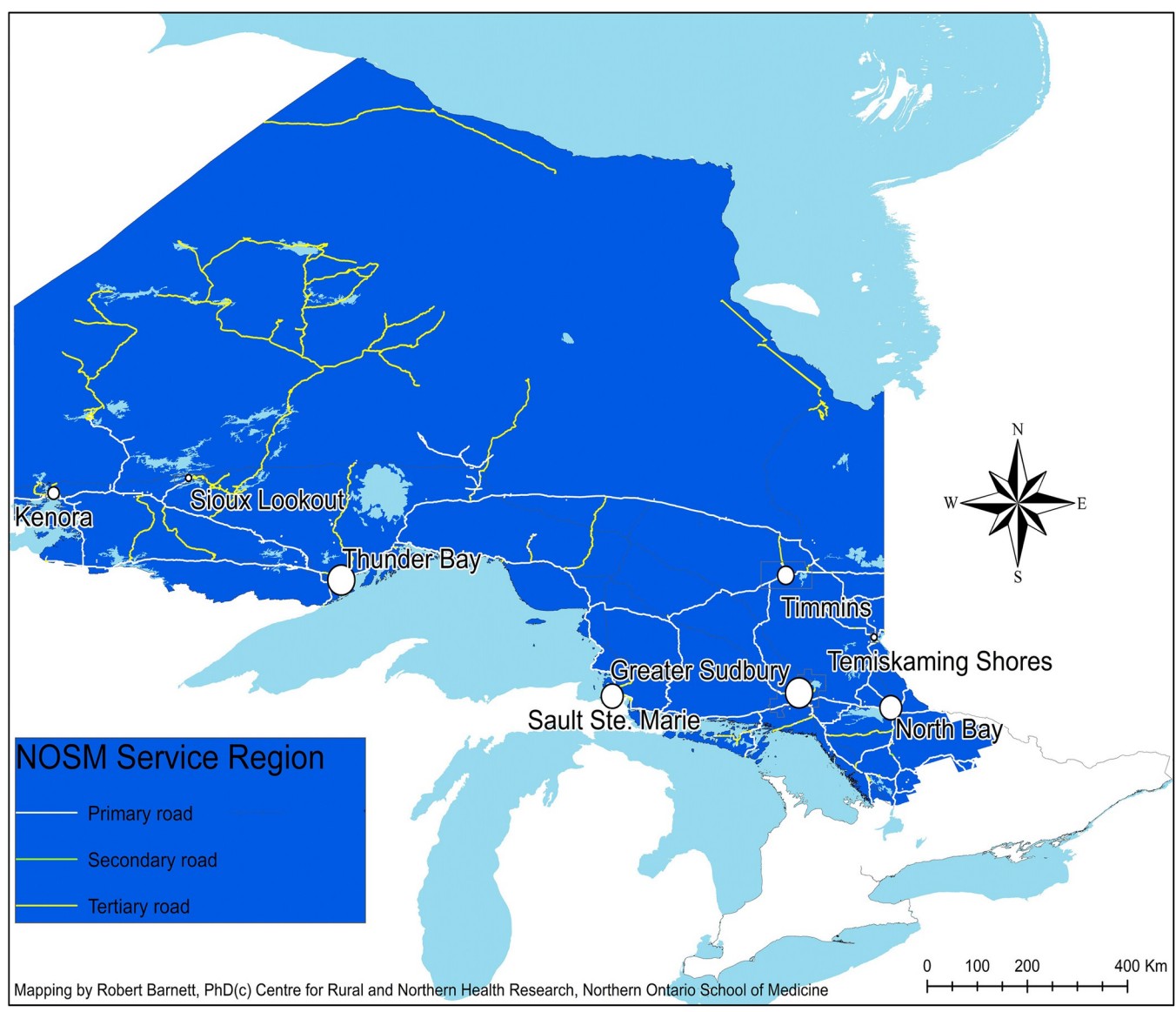

**Fig 1. Map showing the major communities and transportation network in the service region of the Northern Ontario School of Medicine.**

Hometown, self-reported on CRaNHR surveys and medical school applications, was defined as the CSD in which the doctor lived for 9 or more years from birth to 18 years of age. Primary practice location in November 2019 was obtained from publicly available information held by medical licensing and regulatory authorities in the provinces and territories of Canada.

We categorised doctors by their education path. Paths included: (1) NOSM UG and NOSM PG; (2) NOSM UG and Other medical school PG (i.e., NOSM UG only); and (3) Other school UG and NOSM PG (i.e., NOSM PG only). Doctors were also categorised into broad specialty groups: (1) family medicine specialty with PG training offered at NOSM or at other Canadian medical schools; and (2) generalist specialties (i.e., non-family practice generalist specialties) with PG training offered at NOSM or at other schools; and (3) other specialties with PG training offered at other schools but not at NOSM (S1 Table). In November 2019, Family Physicians (FPs) and general specialists were in practice for 1–8 years and 1–6 years, respectively.

A first set of independent variables were based on previous research on NOSM learners [20–23] and included: UG-PG path; specialty group; northern Ontario hometown; rural hometown; gender (male or female); and age in 2019 (z-score). A second set of variables were only used in exploratory analyses because more data were missing or because some variables had data combined from different sources. The second data set included: return-of-service contract; Indigenous status; French language ability (all as yes or no); and year commencing medical school (2005–2011). Return-of-service contract length (years) was self-reported on CRaNHR surveys and therefore data were missing for those who did not complete the surveys. Indigenous status and French language ability were compiled from multiple sources, which likely introduced an unknown amount of error. Sources of data on Indigenous status and French language ability included: NOSM's admission records (requiring proof of status or language ability); [27] NOSM administrative records (self-reported); and CRaNHR surveys (self-reported). Self-reported ability to provide French-language service was also obtained from the College of Physicians and Surgeons of Ontario (CPSO) registration data and was therefore only available for doctors who practised in Ontario and who stated that they were willing and able to practice in French.

We used binary logistic regression to predict northern Ontario practice or rural Canada practice from the first variable set. Cases with missing data needed for the logistic regression models were excluded from analyses. We tested separate models for each specialty group because of the expected relationship between specialty group and practice community population size [28, 29]. We tested model assumptions (e.g., sample size, linearity in the logit, multi-collinearity) and assessed model fit with Nagelkerke's R-squared, Hosmer and Lemeshow's goodness-of-fit test, and the area under the curve of one minus specificity versus sensitivity [30]. To minimize the risk of deleting salient variables too soon from the model, we removed variables during backwards elimination when p>0.20 [31].

We used cross tabulations with unadjusted odds ratio (OR) and exact tests to probe significant associations identified by logistic regression [32, 33]. No missing data were imputed and no adjustments were made for multiple tests. We used IBM SPSS for Windows version 26 for all statistical analyses [34].

## Results

The initial dataset included 692 doctors from which we removed 97 IMGs, 77 doctors with less than 1 year of practice, and 27 doctors who came to NOSM only for their Family Medicine PGY3, which yielded 491 doctors (S2 Table). Fifty-six of these doctors were missing data needed for logistic regression and therefore analyses were conducted with records from 435 doctors.

**Table 1.**

**Table 1a. Demographic characteristics of 435 NOSM-trained doctors.***

| Demographic | n (%) except as noted |
|---|---|
| Age at PG entry | 30 (5.3) mean (standard deviation) |
| | 29 (24–56) median (minimum–maximum) |
| Female (remainder were Male) | 295 (67.8) |
| Indigenous Person | 37 (8.9) (missing data for 17 cases) |
| French language ability | 166 (36.4) (missing data for 1 case) |

**Table 1b. Hometown: Rural/Urban by Region (n (% of total)).**

| Rural/Urban | Northern Ontario | Other region | Rural/Urban Total |
|---|---|---|---|
| Rural | 101 (23.2) | 37 (8.5) | **138 (31.7)** |
| Urban | 216 (49.7) | 81 (18.6) | **297 (68.3)** |
| **Region Total** | **317 (72.9)** | **118 (27.1)** | **435 (100)** |

**Table 1c. Medical education path and specialty group.**

| Undergraduate medical education school-Postgraduate residency training school | n (%) | |
|---|---|---|
| NOSM-NOSM | 146 (33.6) | |
| Other-NOSM | 85 (19.5) | |
| NOSM-Other | 204 (46.9) | |
| | **435 (100)** | |
| **Specialty Group †** | **n (%)** | |
| Family Medicine (at NOSM or other schools) | 334 (76.8) | |
| Generalist Specialties (at NOSM or other schools) | 62 (14.3) | |
| All other specialties (only at other schools) | 39 (9.0) | |
| | **435 (100)** | |

Note: NOSM = Northern Ontario School of Medicine, Other = Other Canadian medical school.

* Refer to Methods for definitions of Indigenous Person, French language ability, rural, urban, and northern Ontario.

† Refer to S1 Table for a list of specialties by group.

Doctors' average age at entry to residency was 30 years and 68% were female (295/435) (**Table 1A**). Approximately 9% were Indigenous (37/418) and 38% (166/434) had French-language ability. Over two-thirds (297/435 = 68%) reported an urban hometown and 73% (317/435) were from northern Ontario (**Table 1B**). Approximately 23% (92/392) of doctors reported that they had signed a return-of-service contract lasting 1–6 years. Approximately 34% (146/435) stayed at NOSM for both UG and PG, while 20% (85/435) completed only their PG training at NOSM and 47% (204/435) completed only their UG at NOSM (**Table 1C**). Approximately 77% (334/435) were licensed in family medicine, with 14% (62/435) licenced in generalist specialties, and 9% (39/435) licensed in other specialties (**Table 1C**).

To compare practice location among FPs, other generalist specialists, and other specialists, we extracted counts from **Table 2**, and used exact tests to look for differences among mutually exclusive categories. For example, collapsing rural and urban practice locations and looking only at region, significantly more FPs (201/334 = 60%) were practising in northern Ontario than generalist specialists (23/62 = 37%) or other specialists (9/39 = 23%) (exact test, p<0.001) (**Table 2**). Similarly, collapsing across regions, significantly more FPs (93/334 = 28%) practised in rural Canada than generalist specialists (3/62 = 5%) or other specialists (1/39 = 3%) (exact test, p<0.001).

**Table 2. Physicians' practice location: rural/urban by region by specialty group.**

|  | Northern Ontario | Other Northern Region in Canada * | Not Northern Canada † | Total |
|---|---|---|---|---|
| Family Medicine (at NOSM or at other medical schools) |  |  |  |  |
| Rural | 59 | 0 | 34 | **93** |
| Urban | 142 | 3 | 96 | **241** |
| **Subtotal** | **201** | **3** | **130** | 334 |
| Other Generalist Specialties (at NOSM or at other medical schools) |  |  |  |  |
| Rural | 2 | 0 | 1 | **3** |
| Urban | 21 | 0 | 38 | **59** |
| **Subtotal** | **23** | **0** | **39** | 62 |
| All other Specialties (only at other medical schools) |  |  |  |  |
| Rural | 0 | 0 | 1 | **1** |
| Urban | 9 | 0 | 29 | **38** |
| **Subtotal** | **9** | **0** | **30** | 39 |
| All Specialties |  |  |  |  |
| Rural | 61 | 0 | 36 | **97** |
| Urban | 172 | 3 | 163 | **338** |
| **Grand Total** | **233** | **3** | **199** | 435 |

Note: NOSM = Northern Ontario School of Medicine, Other school = Other medical school (not NOSM).

* Other Northern region in Canada as defined by each of the remaining provinces plus any location in Nunavut, Northwest Territories, or Yukon Territories.

† Not in Northern Canada includes international locations.

## Predicting northern Ontario practice

For FPs, the logistic regression model that predicted practice in northern Ontario versus all other regions used UG/PG path, hometown region, gender, and age in 2019. The full model improved the total percentage correctly predicted from 60% (constant only) to 81%, and diagnostics indicated a robust model (**Table 3**). Specifically, the full model correctly predicted 78% (157/201) and 84% (112/133) of FPs practising inside and outside northern Ontario, respectively.

For FPs practising in northern Ontario, completing UG and PG at NOSM had a statistically significant (p<0.001) adjusted OR (aOR) of 45.8 (95% confidence interval (CI) = 20.3–103.0)

**Table 3. Logistic regression model to predict northern Ontario practice location of 334 family doctors\*.**

| Variables in the model |  |  |  |  |  |  | 95% CI for $e^\beta$ | |
|---|---|---|---|---|---|---|---|---|
|  | β | SE(β) | Wald | df | p-value | $e^\beta$ | Lower | Upper |
| NOSM UG/Other PG (reference) |  |  | 92.1 | 2 | <0.001 |  |  |  |
| NOSM UG/NOSM PG | 3.82 | 0.41 | 85.1 | 1 | <0.001 | 45.8 | 20.3 | 103.2 |
| Other UG/NOSM PG | 2.71 | 0.47 | 33.7 | 1 | <0.001 | 15.0 | 6.0 | 37.5 |
| Northern Ontario Hometown | 1.67 | 0.43 | 15.4 | 1 | <0.001 | 5.3 | 2.3 | 12.2 |
| Female Gender | 0.51 | 0.31 | 2.6 | 1 | 0.11 | 1.7 | 0.9 | 3.1 |
| Doctor's Age in 2019 (z-score) | -0.06 | 0.16 | 0.2 | 1 | 0.69 | 0.9 | 0.7 | 1.3 |
| Constant | -2.81 | 0.47 | 23.1 | 1 | <0.001 | 0.1 |  |  |

Note: β = model coefficient, CI = confidence interval, df = degrees of freedom, $e^\beta$ = odds ratio adjusted for all variables in the model, NOSM = Northern Ontario School of Medicine, Other = other Canadian medical school (not NOSM), p-value = probability, SE = standard error

* Model diagnostics: Nagelkerke $R^2$ = 0.53; Hosmer and Lemeshow goodness-of-fit test p = 0.30; Area under the curve = 0.88 (p<0.001) for a plot of 1-specificity versus sensitivity.

relative to the reference group of FPs who completed only their UG at NOSM (**Table 3**). The aOR for FPs who completed only their PG at NOSM was 15.0 (95% CI = 6.0–37.5). Having a northern Ontario hometown was a significant predictor of a northern Ontario practice location (p<0.001) (aOR = 5.3, 95% CI = 2.3–12.2). Gender and age were not significant predictors (p≥0.10). Backwards elimination modelling for 294 FPs with data for all primary and secondary variables did not identify any other predictors. Robust logistic regression models, including those with fewer variables, could not be built for non-family medicine specialties for northern Ontario nor for rural Canada practice location outcome.

Detailed examination of the interaction between northern Ontario hometown and UG-PG path found that FPs with a northern Ontario hometown had statistically significant (p<0.04) unadjusted ORs of 4.9–5.8 of practising in northern Ontario for all UG-PG paths (**S3 Table**). For instance, 94% (116/123) of FPs who completed their UG and PG at NOSM and had a northern Ontario hometown were practising in northern Ontario, which was higher than any other combination of UG-PG path and specialty group (**S3 Table**). There was no evidence of a similar interaction between UG-PG path and a northern Ontario hometown (p≥0.16) for any non-family medicine specialty group.

## Predicting rural Canada practice

Using logistic regression to predict rural practice location was less effective. For FPs, the logistic regression model that predicted practice in a rural versus urban community minimally improved the total percentage correctly predicted from 72% (constant only) to 73%, with diagnostics suggesting a somewhat robust model (**Table 4**). The model correctly predicted FPs in urban practice (235/241 = 98%) but was a poor predictor of rural practice (8/93 = 9%). Completing only a PG at NOSM was associated with a statistically significant (p = 0.05) aOR of 2.0 (95% CI: 1.0–3.9) relative to the reference group of doctors completing only a UG at NOSM. For FPs, having a rural hometown was a statistically significant predictor of a rural practice location (p<0.01) (aOR = 2.3, 95% CI = 1.3–3.8) as was age (p = 0.01, aOR = 1.4, 95% CI = 1.1–1.8). Approximately 39% (41/106) of FPs with a rural background practised in rural Canada, compared with 23% (52/228) FPs without a rural background (**S4 Table**). Gender was not a significant predictor and backwards elimination modelling conducted on 294 FPs with additional data did not identify any new variables.

Detailed examination of the interaction between rural hometown and UG-PG path found that having a rural hometown had a statistically significant (p<0.001) unadjusted OR of 5.5

**Table 4. Logistic regression model to predict rural practice location of 334 family doctors***.

| Variables in the model | | | | | | | 95% CI for $e^{\beta}$ | |
|---|---|---|---|---|---|---|---|---|
| | B | SE(β) | Wald | df | p-value | $e^{\beta}$ | Lower | Upper |
| NOSM UG/Other PG (reference) | | | 5.3 | 2 | 0.07 | | | |
| NOSM UG/NOSM PG | -0.1 | 0.3 | <0.1 | 1 | 0.82 | 0.9 | 0.5 | 1.7 |
| Other UG/NOSM PG | 0.7 | 0.3 | 3.9 | 1 | 0.05 | 2.0 | 1.0 | 3.9 |
| Rural Hometown | 0.8 | 0.3 | 9.2 | 1 | 0.003 | 2.3 | 1.3 | 3.8 |
| Female Gender | 0.05 | 0.3 | <0.1 | 1 | 0.85 | 1.1 | 0.6 | 1.8 |
| Doctor's Age in 2019 (z-score) | 0.3 | 0.1 | 7.2 | 1 | 0.01 | 1.4 | 1.1 | 1.8 |
| Constant | -1.4 | 0.3 | 28.2 | 1 | <0.001 | 0.2 | | |

Note: β = model coefficient, CI = confidence interval, df = degrees of freedom, $e^{\beta}$ = odds ratio adjusted for all variables in the model, NOSM = Northern Ontario School of Medicine, Other = other Canadian medical school (not NOSM), p-value = probability, SE = standard error

* Model diagnostics: Nagelkerke $R^2$ = 0.08; Hosmer and Lemeshow goodness-of-fit test p = 0.20; Area under the curve = 0.65 (p<0.001) for a plot of 1-specificity versus sensitivity.

(95% CI = 2.4–12.6) for FPs in rural practice, but only for the NOSM UG/NOSM PG path (**S4 Table**). For generalist specialists, significantly more doctors with a rural hometown practised in rural Canada (3/17 = 18%) compared with doctors from an urban hometown (0/45 = 0%) (exact test, p = 0.02). There was no similar associations for other specialists (p = 0.38).

## Discussion

Most fully qualified NOSM-trained doctors (233/435 = 54%) practised in northern Ontario in 2019 and this percentage as well as the magnitude of the adjusted odds ratio (aOR) depended on the specialty group, UG-PG path, hometown region, and the interaction among these factors. For instance, FPs who completed both UG and PG at NOSM had highest magnitude aOR of practising in northern Ontario relative to those FPs who only completed their UG at NOSM. The aOR of 46 was three-times the magnitude of the aOR for completing just the PG at NOSM and almost nine-times the magnitude for having a northern Ontario background. As an example of the interaction, the sub-group of FPs who completed their UG and PG at NOSM and had a northern Ontario hometown had the highest percentage for practicing in northern Ontario (116/123 = 94%). For FPs, selecting qualified students from the region and providing both UG and PG training in the region resulted in the overwhelming majority establishing their independent practice in the region. For general specialists, the limited data suggested similar, albeit smaller and non-significant associations. Results for other specialists were inconclusive.

Our study supported the robust positive association between postgraduate training location and practice region found by previous researchers, with FPs or general practitioners more likely to stay in the same region than other specialists [35–38]. Doctors with a NOSM UG and a NOSM PG showed the strongest association: 92% (128/139) of FPs and 57% (4/7) (non FP) generalist specialists in this UG-PG path practised in northern Ontario (overall: 132/146 = 90%). These values compare favourably with published results from the Faculty of Medicine at the Memorial University of Newfoundland (MUN), which is the sole medical school in the province of Newfoundland and Labrador—a region of Canada that is also underdeveloped and underserved [29]. The overall percentage for FPs with a NOSM UG and NOSM PG was nearly twice that of the 49% (508/1047) of doctors who graduated 1973–2008 from MUN, did at least some of their PG training at MUN and were practising in MUN's service region in 2014 [29].

Overall, 22% (97/435) of NOSM-trained doctors practised in rural Canada and this was almost three-times higher than the 7.7% (6,994/91,372) of all physicians practising in rural Canada in 2019 [35]. Again, FPs had the highest percentage (93/334 = 28%), which is just over two-times the percentage of all FPs practising in rural Canada (6,022/46,131 = 13%) [35]. For FPs and general specialists, having a rural background increased the odds of practicing in a rural community.

The present study replicated the positive association between a rural hometown and FP rural practice location that is strongly supported by research in many countries [39–41]. However, rural hometown, despite the significant bivariate association, did not predict rural practice location: only 9% (8/93) of the FPs known to practise in rural Canada were correctly identified by the logistic regression model. Our study found that 28% (36/139) of FPs and 14% (1/7) non-FP generalist specialists who completed UG and PG at NOSM practised in rural Canada in 2019 (overall: 37/146 = 25%). Again, the overall percentage was nearly twice that of the 14% (84/581) of doctors who completed their UG at MUN, completed some or all of their PG training at MUN, and had a rural Canada practice in 2014 [42]. For MUN graduates without any MUN PG training, only 8% (41/515) practised in rural Canada. In comparison, our

study on NOSM graduates who completed their PG at another school showed that 27% (34/127) of FPs and 2% (1/42) non-FP generalist specialists practised in rural Canada in 2019 (overall: 35/169 = 21%, almost 3-times MUN's percentage). Relative to other UG-PG paths, the combination of a NOSM UG and a NOSM PG may enable physicians with a rural background to practise in rural communities.

Our study did not find any significant association between practice location and learners' gender, French language ability, Indigenous status, or contractual obligation. Several Canadian studies have found no evidence of an association with gender [29, 43, 44] and some evidence for a negative association between first practice location in the region and visible minority status [44]. In addition, proximity to the hometown of the doctor's spouse or partner may be a mitigating factor—a factor that is often acknowledged, but not always measured [45].

The main limitation of this study was the focus on doctors trained at one medical school with only within-school comparison groups. Other limitations may have arisen from simplifying assumptions in the analyses. For example, the nested data structure (e.g., doctors nested within graduation year) was ignored, which may have underestimated standard errors [36]. In addition, broadly categorising UG-PG path and specialty may mask some differences among medical school-specialty combinations. However, our approach was reasonable given small sample sizes for some paths, specialties, and outcomes.

The attractiveness of the hometown region and not just the size of the community in determining where doctors will practise has been previously reported in the literature [37, 38, 46, 47], though its importance is not always well-recognised. For instance, historically much has been made of the positive association between rural hometown and rural practice, without consideration of whether doctors were returning to practice in other communities in their hometown region.

Evidence for a positive interaction between hometown region or rurality, training region or rurality, and practice region or rurality is building. A 2020 review cited four studies with variable evidence for the combined effect of a rural background and rural training on rural practice [48]. Published in 2021, a cross-sectional study of Australian doctors found a strong positive predictive relationship among having the same rural region for their hometown, rural region training location during part of their UG, and rural region of practice in 2017 [49]. There was a consistent dose-response effect between years in rural hometown and practice in the same rural region as well as between training duration in a rural region and practice in the same rural region. Other recent work supports the positive association among hometown, training and practice location [50].

It is also worth noting that while most rural areas are underserved, not all rural areas can support a full-time doctor, especially a doctor with more specialized training. This problem is particularly acute in rural and northern areas of Canada with vast distances between sparsely populated communities. In these very sparsely populated regions, locating practices in nearby urban centres may be the only viable option for many doctors, regardless of specialty.

Recent increases in the use of virtual care in response to restrictions on in-person consultations imposed during the COVID-19 pandemic demonstrate risks and benefits of virtual care for underserved areas [51, 52] and this has implications on the usefulness of our findings. Although virtual care can fill an immediate need for medical services in underserved regions, we suggest that relying on virtual care may be used, wrongly in our view, to justify reduced investment in local services and facilities, thereby further exacerbating inequalities in under-resourced areas. Reliance on virtual care services may also impede workforce recruitment and retention efforts in underserved areas.

However, if virtual care is organized regionally, then this may facilitate opportunities for doctors to locate their practice in one underserved community while providing virtual care

and visiting services for other, adjacent underserved communities. This would provide an adequate population size upon which to base group practices. The key is to ensure that medical practices are physically located in underserved areas, rather than located hundreds of kilometres away and delivering only virtual care services to these areas. Regardless of the evolving role of virtual care, there is still the need to select students from economically deprived areas and to train UG and PG medical students in these areas so as to provide the necessary skills to care for these patients and to encourage physicians to locate their practices in these areas.

Previous research has shown that former NOSM students and residents consider themselves well prepared for rural generalist practice [53]. Our research found that 92% (128/139) of FPs who completed their UG and PG at NOSM were practising in the underserved region of northern Ontario. Collectively, these studies support the potential of the life-cycle [1, 2] and Rural Generalist Pathway approaches [8]. For non FP specialists, the pathway effect was not as clear with 32% (32/101) of all non-family medicine specialists practising in northern Ontario, and little difference among UG-PG paths, though this results may be due to small numbers.

## Conclusions

Our study has demonstrated the interaction of two main mechanisms by which medical schools can significantly increase the proportion of doctors practising in economically deprived, underserviced regions. The first mechanism is to admit medical students who grow up in these regions—this mechanism had a positive association regardless of medical education path. The second, and more influential factor is to provide undergraduate and postgraduate medical education immersed in the regions, whereby students and trainees are living and learning in the community and in the clinical settings where they are expected to practice in the future. NOSM's use of both mechanisms has enabled a substantial majority of NOSM-trained doctors to practise in the underserved region of northern Ontario, with a sizeable percentage practising in rural communities. Purposeful selection and distributed immersive community engaged education during UG and PG are likely to benefit economically deprived and underserved regions around the world.

## Supporting information

**S1 Table. Specialties by specialty group.**
(DOCX)

**S2 Table. Excluded cases and comparison with included cases.**
(DOCX)

**S3 Table. Northern Ontario–hometown vs practice location.**
(DOCX)

**S4 Table. Rural–hometown vs practice location.**
(DOCX)

## Acknowledgments

We thank the Northern Ontario School of Medicine (NOSM University) medical students and residents who consented to be part of this research. We thank Patrick Timony, Centre for Rural and Northern Health Research (CRaNHR–Laurentian), and numerous research assistants who helped with CRaNHR's Tracking Study of NOSM's students and residents. We also thank Robert Barnett, CRaNHR–Laurentian, who prepared the map that appears as Fig 1. The

opinions expressed in this article are those of the authors and are not necessarily shared by the Ontario Ministry of Health, CRaNHR, Laurentian University, or NOSM University.

## Author Contributions

**Conceptualization:** John C. Hogenbirk, Roger P. Strasser.

**Data curation:** John C. Hogenbirk, Margaret G. French.

**Formal analysis:** John C. Hogenbirk.

**Funding acquisition:** John C. Hogenbirk, Roger P. Strasser.

**Investigation:** John C. Hogenbirk, Roger P. Strasser.

**Methodology:** John C. Hogenbirk, Margaret G. French.

**Project administration:** John C. Hogenbirk, Roger P. Strasser, Margaret G. French.

**Resources:** Roger P. Strasser.

**Supervision:** John C. Hogenbirk, Roger P. Strasser.

**Validation:** John C. Hogenbirk, Roger P. Strasser.

**Writing – original draft:** John C. Hogenbirk.

**Writing – review & editing:** John C. Hogenbirk, Roger P. Strasser, Margaret G. French.

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
