## [Decision Letter · Decision Letter 0]

11 Apr 2022

PONE-D-22-06414Ten years of graduates: a cross-sectional study of the practice location of doctors trained at a socially accountable medical schoolPLOS ONE

Dear,

Thank you for submitting your manuscript to PLOS ONE. After careful consideration, we feel that it has merit but does not fully meet PLOS ONE’s publication criteria as it currently stands. Therefore, we invite you to submit a revised version of the manuscript that addresses the points raised during the review process.

We look forward to receiving your revised manuscript.

Kind regards,

Muhammad Shahzad Aslam, Ph.D.,M.Phil., Pharm-D

Academic Editor

PLOS ONE

Journal Requirements:

[JCH works part time for Northern Ontario School of Medicine (NOSM) as a research tutor in the Family Medicine program. RPS is the Founding Dean Emeritus of NOSM. MGF worked for Centre for Rural and Northern Health Research-Laurentian during preparation of the manuscript and since September 2020 has worked full time for NOSM as an Analyst in the Office of Institutional Intelligence.] 

a) You may seek permission from the original copyright holder of Figure 1 to publish the content specifically under the CC BY 4.0 license.  

Reviewers' comments:

Reviewer's Responses to Questions

**Comments to the Author**

1. Is the manuscript technically sound, and do the data support the conclusions?

Reviewer #1: Yes

Reviewer #2: Yes

2. Has the statistical analysis been performed appropriately and rigorously? 

Reviewer #1: I Don't Know

Reviewer #2: Yes

3. Have the authors made all data underlying the findings in their manuscript fully available?

Reviewer #1: Yes

Reviewer #2: Yes

4. Is the manuscript presented in an intelligible fashion and written in standard English?

Reviewer #1: Yes

Reviewer #2: Yes

5. Review Comments to the Author

Reviewer #1: This is an excellent study and most helpful to those who are interested in learning which factors weigh most heavily toward recruitment of primary care into rural and underserved settings. One additional limitation: I understand that Canada manages their residency slots, and this helps produce a higher percentage of family physicians. It may be worth adding this since not all countries will have this in place.

This certainly adds to the literature on the need for rural/underserved exposure during medical school. If not referenced, consider reviewing some of the recent literature out of Australia that also supports this.

Reviewer #2: Comments (L = Line)

L51: Put a reference at the end of the paragraph.

L162: Table 1 needs to be re-organized in a better way.

L163: The total calculation should be either in a form of ‘row’ or ‘column’ total, but the authors have calculated the percentages out of the grand total.

L170 and downwards: There are a lot of explanations and definitions that can be mentioned in the methods section rather than putting them under the table.

L191-196: Table 2 is not accepted in this format. The text contains details like p values and percentages that are not present in the table. The authors should calculate a row percentage and should put the percentage (%) between brackets next to each frequency, and in the last column they should mention the p value.

6. PLOS authors have the option to publish the peer review history of their article (what does this mean?). If published, this will include your full peer review and any attached files.

Reviewer #1: No

Reviewer #2: No

---

## [Author Response · Author response to Decision Letter 0]

10 Jun 2022

Please refer to the uploaded file "response to reviewers"

---

## [Decision Letter · Decision Letter 1]

11 Jul 2022

PONE-D-22-06414R1Ten years of graduates: a cross-sectional study of the practice location of doctors trained at a socially accountable medical schoolPLOS ONE

Dear Dr. Hogenbirk,

Thank you for submitting your manuscript to PLOS ONE. After careful consideration, we feel that it has merit but does not fully meet PLOS ONE’s publication criteria as it currently stands. Therefore, we invite you to submit a revised version of the manuscript that addresses the points raised during the review process. Please submit your revised manuscript by Aug 25 2022 11:59PM. If you will need more time than this to complete your revisions, please reply to this message or contact the journal office at plosone@plos.org. Please include the following items when submitting your revised manuscript:A rebuttal letter that responds to each point raised by the academic editor and reviewer(s). You should upload this letter as a separate file labeled 'Response to Reviewers'.A marked-up copy of your manuscript that highlights changes made to the original version. You should upload this as a separate file labeled 'Revised Manuscript with Track Changes'.An unmarked version of your revised paper without tracked changes. You should upload this as a separate file labeled 'Manuscript'.If applicable, we recommend that you deposit your laboratory protocols in protocols.io to enhance the reproducibility of your results. Protocols.io assigns your protocol its own identifier (DOI) so that it can be cited independently in the future. For instructions see: https://journals.plos.org/plosone/s/submission-guidelines#loc-laboratory-protocols. Additionally, PLOS ONE offers an option for publishing peer-reviewed Lab Protocol articles, which describe protocols hosted on protocols.io. Read more information on sharing protocols at https://plos.org/protocols?utm_medium=editorial-email&utm_source=authorletters&utm_campaign=protocols.

We look forward to receiving your revised manuscript.

Kind regards,

Muhammad Shahzad Aslam, Ph.D.,M.Phil., Pharm-D

Academic Editor

PLOS ONE

Journal Requirements:

Additional Editor Comments :

Please complete the corrections according to reviewer comments otherwise the manuscript does not proceed further

Reviewers' comments:

Reviewer's Responses to Questions

**Comments to the Author**

1. If the authors have adequately addressed your comments raised in a previous round of review and you feel that this manuscript is now acceptable for publication, you may indicate that here to bypass the “Comments to the Author” section, enter your conflict of interest statement in the “Confidential to Editor” section, and submit your "Accept" recommendation.

Reviewer #2: (No Response)

2. Is the manuscript technically sound, and do the data support the conclusions?

Reviewer #2: Yes

3. Has the statistical analysis been performed appropriately and rigorously? 

Reviewer #2: Yes

4. Have the authors made all data underlying the findings in their manuscript fully available?

Reviewer #2: Yes

5. Is the manuscript presented in an intelligible fashion and written in standard English?

Reviewer #2: Yes

6. Review Comments to the Author

Reviewer #2: Dear authors: Plz respond positively to my previous note

L191-196: Table 2 is not accepted in this format. The text contains details like p values and percentages that are not present in the table. The authors should calculate a row percentage and should put the percentage (%) between brackets next to each frequency, and in the last column they should mention the p value.

Regards

7. PLOS authors have the option to publish the peer review history of their article (what does this mean?). If published, this will include your full peer review and any attached files.

Reviewer #2: No

---

## [Author Response · Author response to Decision Letter 1]

3 Aug 2022

This response was first submitted and uploaded as a separate file on May 26, 2022.

Re-submitted/uploaded on June 10, 2022 with minor changes.

Re-submitted/uploaded on August 3, 2022 with additional details on the structure of Table 2.

Thank you for the opportunity to reply to the reviewers’ comments. The comments and our responses are documented below.

Reviewer #1.

Abstract: some of the strongest data, in my opinion, is the aOR of 45 for FPs practicing in northern Ontario who completed UG and PG at NOSM vs. rural background aOR = 5 Would recommend adding this to the abstract.

We have revised the abstract to include the adjusted odds ratios and their 95% confidence intervals.

Line 21 “We used a cross-sectional study to measured, …” Is the word “measured” necessary? It first appears to be a typo (e.g. to measure)

Our thanks for identifying these and other typos. 

The abstract was revised to include the above suggestion and the typo was resolved.

Line 47-48 “NOSM selects students from underserved and economically deprived regions of Canada with a focus on northern Ontario.” This can be strengthened with some data, such as “____% of matriculating students at NOSM are from….”

We’ve revised this paragraph to add information on matriculating students.

Line 60 typo “with most of the these”

Fixed, thanks.

Lines 269-270. “Completing both UG and PG at NOSM had the highest aOR of 45.7.” Since there is a lot of data, would recommend qualifying this statement, “Completing both UG and PG at NOSM had the highest aOR of 45.7 for FPs practising in northern Ontario.”

We have revised the relevant sentences to read…” FPs who completed both UG and PG at NOSM had highest magnitude adjusted odds ratio (aOR) of practising in northern Ontario relative to those FPs who only completed their UG at NOSM. This aOR of 46 was three-times the magnitude of the aOR for completing just the PG at NOSM and almost nine-times the magnitude for having a northern Ontario background.”

Line 280-282 “The overall percentage was nearly twice that of the 49% (508/1047) of doctors who graduated 1973 – 2008 from the Memorial University of Newfoundland (MUN) and did at least some of their PG training at MUN and were practising in MUN’s service region in 2014.[27]” Intriguing. A very nice comparator. For those not familiar, what is the rationale for using MUN as a comparator? Are they similarly situated? 

We have added information on how MUN compares with NOSM.

Lines 326-334 seem out of place for this manuscript [“Recent increases in the use of virtual care…”]

Our rationale for including this paragraph is that virtual care offers an alternative to physically locating physicians in regions of need. We argue that, at least, physicians should be trained to care for patients in underserved regions if they are to provide effective virtual care. We also suggest that it is better still to have physicians locate their practice in underserved areas. We have added text to flesh out our arguments.

Line 347 It’s fair to say the following, yes? The second, and more influential factor in our study, is to provide undergraduate and postgraduate medical education immersed in the regions, whereby students and trainees are living and learning in the community and in the clinical settings where they are expected to practice in the future.

We added the suggested clause to this sentence. 

Many are going to be interested in this conclusion because while we recognize the value of admitting students from rural backgrounds into medical school, and we recognize the role of the medical school educational environment, we do not know which is more influential. This study appears to help answer that. 

This is an excellent study and most helpful to those who are interested in learning which factors weigh most heavily toward recruitment of primary care into rural and underserved settings.

We thank the reviewer for these kind comments

One additional limitation: I understand that Canada manages their residency slots, and this helps produce a higher percentage of family physicians. It may be worth adding this since not all countries will have this in place.

We have added some information to the Methods section on medical education in Canada and on residency matching to provide readers with the necessary context.

This certainly adds to the literature on the need for rural/underserved exposure during medical school. If not referenced, consider reviewing some of the recent literature out of Australia that also supports this.

We have added more references including some very recently published studies from Australia. 

Reviewer #2

L51: Put a reference at the end of the paragraph. 

We have revised the paragraph and have included the appropriate reference.

L162: Table 1 needs to be re-organized in a better way. 

Table 1 has been reorganized.

L163: The total calculation should be either in a form of ‘row’ or ‘column’ total, but the authors have calculated the percentages out of the grand total. 

We have calculated percentages in the revised Table 1 to best illustrate the point that we wish to make. We used the percentage of the total in the part of the Table that presents rural or urban hometown vs hometown region because we don’t wish to prioritize rurality (row) over region (column). 

L170 and downwards: There are a lot of explanations and definitions that can be mentioned in the methods section rather than putting them under the table. 

We have added to the existing material in the Methods section and have removed most of the Table notes.

L191-196: Table 2 is not accepted in this format. The text contains details like p values and percentages that are not present in the table. The authors should calculate a row percentage and should put the percentage (%) between brackets next to each frequency, and in the last column they should mention the p value.

[Authors’ reply was updated August 3, 2022, elaborating on the reply that was uploaded May 26 and June 11.]

Table 2 contains detailed data that were collapsed in different ways in the text to test for differences among the three specialty groups in rural-urban practice location or in practice region. We have added text to the manuscript to better describe the specific values that we extract from Table 2, which are used in these exact tests. August 3, 2022 tracked changes in the manuscript are highlighted in yellow.

We deliberately presented only the counts and not percentages in Table 2 to avoid confusion. We suggest that Table 2 would become too cluttered and complex if we were to place the percentages (calculated on column totals within each specialty for one comparison and calculated on row totals within each specialty for the other comparison), plus display p-values and highlight the desired comparisons in Table 2. 

The current structure of Table 2 also helps with data disclosure requirements, by allowing the reader to extract counts and run customized analyses (provided the data that they extract are mutually exclusive and complete). For example, the reader could conduct exact tests within a specialty group (e.g., within the FP group), run a hierarchical log-linear model with specialty group (n=3) by rurality (n=2) by region (n=3), or collapse some categories and run additional tests. 

We also suggest that it is accepted practice to extract a subset of data from a larger table and test pre-defined hypotheses as we have done in our manuscript. The explanatory text added to the manuscript on August 3, 2022, should help clarify our approach and justify the current structure of Table 2.

If there are residual concerns about Table 2, then one alternative is to create two separate and collapsed Tables to reflect the two comparisons and repurpose the detailed Table 2 as a supplement. This alternative seems unnecessarily redundant and hence we prefer to use Table 2 as is, with additional explanation in the text. 

We trust that we have addressed the major concerns expressed by the reviewers and editor(s). Please contact us should you have any further questions.

Respectfully submitted,

John C. Hogenbirk, Roger Strasser, Margaret French

---

## [Decision Letter · Decision Letter 2]

30 Aug 2022

Ten years of graduates: a cross-sectional study of the practice location of doctors trained at a socially accountable medical school

PONE-D-22-06414R2

Dear,

We’re pleased to inform you that your manuscript has been judged scientifically suitable for publication and will be formally accepted for publication once it meets all outstanding technical requirements.

Kind regards,

Muhammad Shahzad Aslam, Ph.D.,M.Phil., Pharm-D

Academic Editor

PLOS ONE

Additional Editor Comments (optional):

Reviewers' comments:

Reviewer's Responses to Questions

**Comments to the Author**

1. If the authors have adequately addressed your comments raised in a previous round of review and you feel that this manuscript is now acceptable for publication, you may indicate that here to bypass the “Comments to the Author” section, enter your conflict of interest statement in the “Confidential to Editor” section, and submit your "Accept" recommendation.

Reviewer #3: All comments have been addressed

2. Is the manuscript technically sound, and do the data support the conclusions?

Reviewer #3: Yes

3. Has the statistical analysis been performed appropriately and rigorously? 

Reviewer #3: Yes

4. Have the authors made all data underlying the findings in their manuscript fully available?

Reviewer #3: Yes

5. Is the manuscript presented in an intelligible fashion and written in standard English?

Reviewer #3: Yes

6. Review Comments to the Author

Reviewer #3: The paper has been improved according to the reviewers' suggestions. Now the manuscript can be accepted for publication.

7. PLOS authors have the option to publish the peer review history of their article (what does this mean?). If published, this will include your full peer review and any attached files.

Reviewer #3: No

---

## [Editor Report · Acceptance letter]

6 Sep 2022

PONE-D-22-06414R2 

Ten years of graduates: a cross-sectional study of the practice location of doctors trained at a socially accountable medical school 

Dear Dr. Hogenbirk:

I'm pleased to inform you that your manuscript has been deemed suitable for publication in PLOS ONE. Congratulations! Your manuscript is now with our production department. 

Kind regards, 

on behalf of

Dr. Muhammad Shahzad Aslam 

Academic Editor

PLOS ONE